# Direct and integrating sampling in terahertz receivers from wafer-scalable InAs nanowires

Kun Peng[1,5], Nicholas Paul Morgan [2,5], Ford M. Wagner[1], Thomas Siday [1], Chelsea Qiushi Xia[1], Didem Dede [2], Victor Boureau [3], Valerio Piazza[2], Anna Fontcuberta i Morral [2,4] ✉ & Michael B. Johnston [1] ✉

Terahertz (THz) radiation will play a pivotal role in wireless communications, sensing, spectroscopy and imaging technologies in the decades to come. THz emitters and receivers should thus be simplified in their design and miniaturized to become a commodity. In this work we demonstrate scalable photoconductive THz receivers based on horizontally-grown InAs nanowires (NWs) embedded in a bow-tie antenna that work at room temperature. The NWs provide a short photoconductivity lifetime while conserving high electron mobility. The large surface-to-volume ratio also ensures low dark current and thus low thermal noise, compared to narrow-bandgap bulk devices. By engineering the NW morphology, the NWs exhibit greatly different photoconductivity lifetimes, enabling the receivers to detect THz photons via both direct and integrating sampling modes. The broadband NW receivers are compatible with gating lasers across the entire range of telecom wavelengths (1.2–1.6 μm) and thus are ideal for inexpensive all-optical fibre-based THz time-domain spectroscopy and imaging systems. The devices are deterministically positioned by lithography and thus scalable to the wafer scale, opening the path for a new generation of commercial THz receivers.

Terahertz (THz) spectroscopy and imaging systems are in high demand for numerous practical applications, including airport security screening[1,2], chemical and materials analysis[3], astronomy[4], quality control of food and agricultural goods[5], non-destructive testing and inspection[6,7], environmental monitoring[8], medical diagnosis and biological sensing[9,10]. The advent of the THz time-domain spectrometer and early developments in the field[11–14] were based on free-space femtosecond lasers, such as Kerr-lens mode-locked Ti:sapphire lasers. These free-space THz time-domain spectrometers typically require long optical path lengths, multiple optical elements (e.g. lenses and mirrors) and complex optical arrangements. More recently, advancements in femtosecond fibre lasers in the range of 1–2 μm have enabled more portable, optical-fibre-coupled THz time-domain spectroscopy and imaging systems. Common examples are the ultrafast Yb-fibre lasers at ~1.03 μm[15,16], ultrafast Er-fibre lasers at ~1.55 μm[17] and ultrafast Tm-fibre or Ho-fibre lasers at ~2 μm[18]. This shift of optics from free space to optical fibres has led to more cost-effective and portable THz spectrometers that can operate in extreme environmental conditions with excellent stability. Hence, fibre-coupled THz spectrometers make THz spectroscopy and imaging technologies more accessible and flexible, which is likely to enable new applications, for example in manufacturing quality control[19,20] and THz endoscopic imaging for cancer detection[21,22].

Commercial fibre-coupled THz time-domain spectrometers have been recently released by a range of manufacturers. THz emitter and receiver technologies that are appropriate for operation with the 1–2 μm wavelength (telecom) band of fibre femtosecond lasers include spintronic THz emitters[23–25], electro-optic polymer THz

[1]Department of Physics, University of Oxford, Clarendon Laboratory, Parks Road, Oxford OX1 3PU, UK. [2]Laboratory of Semiconductor Materials, Institute of Materials, EPFL, 1015 Lausanne, Switzerland. [3]Interdisciplinary Centre for Electron Microscopy, EPFL, 1015 Lausanne, Switzerland. [4]Laboratory of Semiconductor Materials, Institute of Physics, EPFL, 1015 Lausanne, Switzerland. [5]These authors contributed equally: Kun Peng, Nicholas Paul Morgan. ✉e-mail: anna.fontcuberta-morral@epfl.ch; michael.johnston@physics.ox.ac.uk

emitter-receiver pairs[26,27] and photoconductive antennas[28]. Amongst them, photoconductive antennas stand out for currently available commercial all-fibre THz systems owing to their simplicity, compactness, and ease of direct coupling to fibre optics. Existing research focuses on the development of 1.55 μm-switched switched photoconductive materials with low dark current such as InGaAs[29,30], InGaAs/InAlAs heterostructures[31,32], InGaAsP[30,33,34] and GaBiAs[35,36].

In contrast to ternary and quaternary alloys, binary compound semiconductors are particularly stable against composition inhomogeneity but lack bandgap tunability. InAs, as a narrow bandgap binary semiconductor is compatible with a wide range of fibre lasers, even for ultrathin devices. However, bulk InAs exhibits high thermal noise and a slow photoresponse (under on-off switching) in devices owing to their high dark current and relatively long charge-carrier lifetime (~nanoseconds[37]), leading to a high noise level that is not suitable for photoconductive THz detection. In stark contrast, InAs nanowires (NWs) retain a high electron mobility and maintain broadband photosensitivity while exhibiting low dark current, thus resulting in meaningful photoconductive THz receivers. It is worth noting that for bulk semiconductor materials it is often best to choose the bandgap to be near resonant with the excitation laser's photon energy in order to minimise dark conductivity from thermal intrinsic charge generation, Instead, photoconductive receivers based on nanoscale and ultra-thin film semiconductors require photon energies significantly larger than the bandgap energy to achieve sufficient absorption to enable switching, making InAs an excellent choice[38]. Furthermore, the geometry of NWs can further aid absorption[39,40], allowing constraints to be relaxed on laser wavelengths suitable for achieving switching of photoconductive devices. As opposed to conventional bulk and thin-film based photoconductive THz receivers, NWs offer excellent polarisation sensitivity to THz radiation[39], combined with negligible electrical cross-talk between detector elements and lower dark current, making them ideal for THz receiver arrays or multi-channel receivers[41]. NWs also possess ultrashort charge-carrier lifetime (~picoseconds or sub-picoseconds) owing to their intrinsic high surface area and crystal faceting, where surface recombination limits the carrier lifetime[42]. The crystallographic nature of the NW's lateral surface therefore affects its light emission, photodetection performance[43–45] and chemical activity[46]. For example, Azimi et al. demonstrated that taper-free {110} side facets of GaAs NWs[44,45] exhibit one order of magnitude lower surface recombination velocity ($3.5 \times 10^4$ cm s$^{-1}$) than bare GaAs NWs with tapered {110}[45] and {112}[47] facets. The finding enabled low-temperature lasing without NW surface passivation. We have recently reported the growth of wafer-scalable horizontal NWs where faceting and cross-sectional shape can be engineered as a function of growth parameters and pattern geometry[48,49]. This provides additional tools to tune surface recombination dynamics. Similarly, it has been found that charge-carrier concentration inside the NWs can be tuned by doping[50,51] and field effects[52]. These findings provide routes for further material functionalisation.

In this work, we demonstrate room-temperature photoconductive-type InAs-NW THz receivers. This represents an original use of scalable, horizontal InAs NWs as functional components in a real device. To the best of our knowledge, neither of these achievements have been previously reported. Our InAs NWs are grown horizontally on GaAs nanoridges via selective-area molecular beam epitaxy (MBE), an approach which physically separates the active NW component from impurities present on the substrate surface due to previous process steps[51]. Bow-tie antennas are deposited to increase light absorption in the horizontal configuration, as well as increase the efficiency in collecting the radiation[53,54]. Device fabrication of traditional vertical free-standing NWs[55] requires a time-consuming transfer and alignment procedure by nano-manipulation[56,57] or transfer printing[41,58]. In contrast, this method is inherently wafer scalable[59], as the horizontal NW arrays can be straightforwardly processed into receivers on the as-grown substrate. This largely reduces the device fabrication complexity, hence promising a practical route towards kilo-pixel arrays of NW THz receivers integrated in focal-plane-array THz cameras for high-speed and real-time THz time-domain spectroscopic imaging. The InAs NW receivers presented here have the potential to outperform commercially available THz cameras[60] (e.g. focal plane III–V high-electron-mobility transistor array, focal plane silicon CMOS circuit array, focal plane microbolometer array and focal plane pyroelectric device array), which either have a slow response speed (in the range of a few to hundreds of milliseconds) or have a limited spectral response range (below 1 THz) while only measuring signal intensity. Our receivers allow phase-sensitive coherent detection with a wide choice in excitation laser sources (that cover a wavelength range of at least 1.2–1.6 μm) and have a broad THz detection bandwidth of 3 THz. This represents the widest detection bandwidth achieved by NW-based photoconductive THz receivers[41,61–63]. Additionally, the selective area growth of horizontal InAs NWs provides highly tuneable photoconductivity lifetimes, which enables THz detection via both direct and integrating sampling of the pulse. Altogether, these results lay the foundation for the development of next-generation broadband high-speed THz cameras which have the potential to be a break-through technology boosting telecom-laser-based, all-fibre THz spectroscopy and imaging systems. The fabrication scalability and properties tunability of the horizontal InAs NWs also promise room for further functionalisation via pattern engineering and polarisation-sensitive detection[41].

## Results

### Fabrication of photoconductive InAs nanowire receivers

Horizontal InAs NWs were grown on top of GaAs nanoridges by selective-area MBE on semi-insulating (100) GaAs substrates. The substrate was prepared using electron-beam lithography to define the position, length and number of NWs as described in refs. 48,49. After the growth of GaAs nanoridges, performed as in ref. 48, the InAs NWs were grown conformally on the facets of the GaAs nanoridges. The GaAs nanoridges therefore act as a growth template and the NWs adopt the same zinc-blende structure as the nanoridges. More details of the growth conditions and parameters can be found in Methods.

The final morphology of the InAs NW depends on the initial shape of the nanoridges, which can be engineered by modifying the array arrangement, dimensions and growth conditions[48,49]. The sketches in Fig. 1a, b illustrate the effect of the GaAs nanoridge shape on the final InAs NW morphology. Figure 1c, d corresponds to representative cross-sectional scanning transmission electron micrographs (STEM) of such NWs. The InAs NW in Fig. 1c exhibits a multitude of facets belonging to the {311} to {111} plane families, while in Fig. 1d only {111} facets are present. For simplicity reasons we will call these NWs as multi- and single- faceted NWs, respectively. The idea of working on these two kinds of NWs is to capitalise on the effect of nanoscale roughness and faceting in NWs on carrier recombination.

Whether a NW is multi-faceted or single faceted depends on the stage of development of the underlying GaAs nanoridge at the moment that the growth switches from GaAs to InAs. If the GaAs nanoridge is underdeveloped, a multi-faceted NW will arise. If the GaAs nanoridge is fully developed, a single-faceted NW will result. The stage of development of the GaAs nanoridge is a function of its local growth rate, which in turn is dependent on the presence and proximity of neighbouring nanoridges, and the growth time, as described in ref. 48. This allows for precise control over whether a NW is single- or multi-faceted.

In the particular case of this work, facet engineering was achieved by varying the number of NWs in a parallel array. This modifies the local growth conditions and allows one to obtain different morphologies in one same chip. Figure 1e–g corresponds to scanning electron micrographs (SEM) of the bow-tie antenna devices produced with an

increasing number of NWs, respectively 1, 5 and 20. Under these conditions, the underdeveloped GaAs nanoridges with multi-faceted InAs NWs are found in the single NW configuration (in contrast to the single-faceted InAs NWs that are found in multiple-parallel-NW configuration). This is due to a lower local density of precursors during the growth as explained in refs. 48,49. Hereafter the devices will be referred to as 1NW, 5NW and 20NW. By adjusting the growth time, it would be equally possible to create parallel arrays of multi-faceted NWs or isolated single-faceted NWs, though this is not achievable on a single-growth substrate. For all the devices, the InAs NWs were grown in openings with similar feature size (the nominal width is 200 nm and the nominal length varies from 6.75 to 40 μm depending on the device configuration), and under the same growth conditions (III–V ratio, temperature, growth time; see Methods).

After NW growth by MBE, bowtie-shaped gold pads were deposited (see Methods for details), functioning as both electrode and antenna to collect and concentrate the incident THz radiation on the NW(s). Bowtie antennas, consisting of two triangles faced tip-to-tip and separated by a small opening gap, were chosen for the receiver, as they can create a highly enhanced local electric field in the gap region (where NWs are located) and provide broadband response in the infrared (IR) and THz regime owing to the scalability of the shape. The tips of the antennas were intentionally rounded to create a relatively uniform local field in the gap region. This modification improved detection sensitivity and was particularly beneficial for multiple-parallel-NW receivers. Details of the bowtie antenna structure used in our experiments are provided in the Supplementary Information (see Fig. S1). For multiple-parallel-NW receivers, the NWs were 40 μm in length with 1 μm spacing between neighbouring NWs, where the antenna gap was 1 μm. For single-NW receivers, the length of the NWs varied depending on the antenna gap in order to maintain a 3-μm overlap between the antenna and the NW on each end. Data presented here corresponds to a 1-μm antenna gap unless otherwise stated.

Before investigating the THz response, we characterised the room-temperature spectral photocurrent response of the receivers.

We measured a broadband response ranging from 500 nm to 3 μm (see Supplementary Information, Fig. S2). The cut-off wavelength of 3 μm is in agreement with the band-edge of zinc blende InAs at room temperature[64,65].

## THz responses of the photoconductive InAs nanowire receivers

Next, we characterised our NW devices for THz sampling. The NWs were exposed to single-cycle electromagnetic pulses, produced by a broadband spintronic THz emitter[25]. The sampling was achieved by optically switching on the NWs by a 35-fs laser pulse with tunability from 1.2 to 1.55 μm. Both the THz pulse and the ultrafast infra-red (IR) laser pulse were focused on the gap region of the receiver as sketched in Fig. 2a. More details are provided in the Methods and Supplementary Information (see Fig. S3).

The IR laser gate pulse generates electron-hole pairs in the semiconductor NW(s), while the THz pulse induces a bias between electrodes that drives a current transient through a closed external circuit (proportional to the THz field). Current is recorded as a function of time delay τ between the THz pulse and the gate pulse allowing the incident THz electric field $E_{(THz)}(t)$ to be recovered according to[66]

$$I(\tau) \propto \int_{-\infty}^{+\infty} E_{THz}(t)\sigma(t - \tau)dt \qquad (1)$$

where $\sigma(\tau)$ is the time-dependant photoconductivity of the detection material.

Figure 2a illustrates two limiting cases of the Eq. (1), Case 1 (direct sampling): the photoconductivity lifetime is much shorter than the duration of the THz pulse (<<1 ps), so the receiver measures only the point of the THz pulse that temporally overlaps with the ultrashort gate pulse. In this scenario, the conductivity $\sigma(\tau)$ of the detection material can be considered as a $\delta$ function. Hence, Eq. (1) can be simplified as

$$I(\tau) \propto E_{(THz)}(t = \tau) \qquad (2)$$

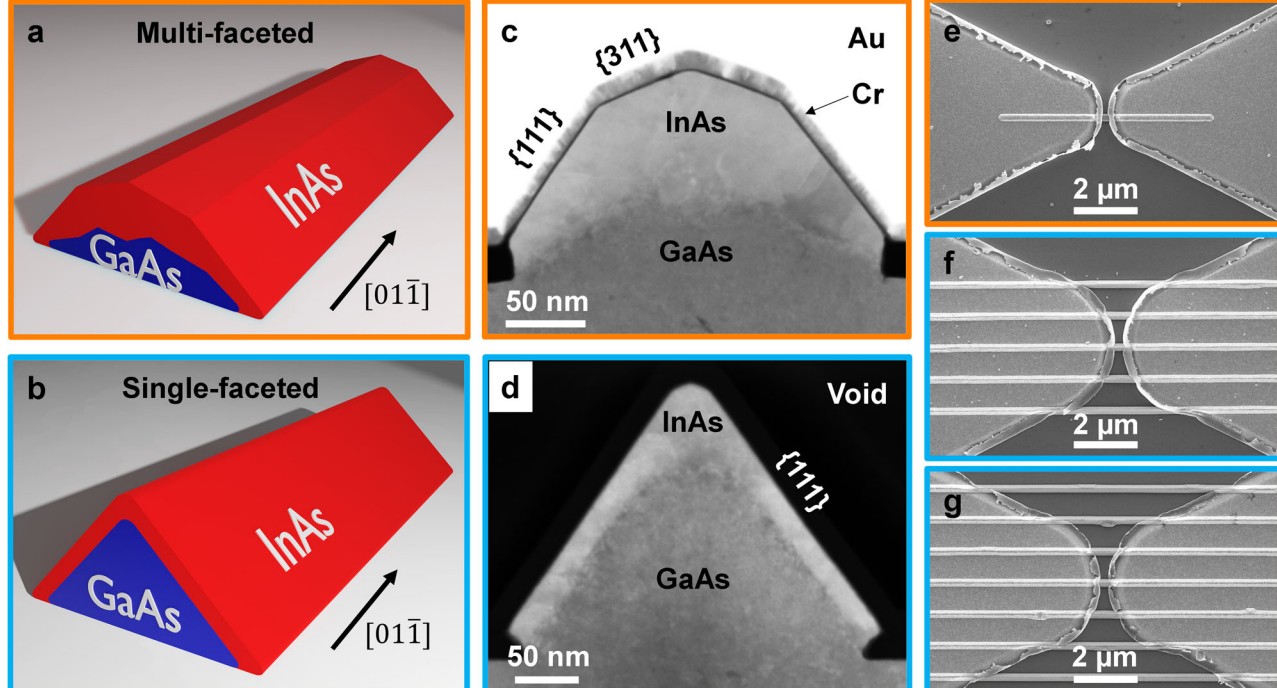

**Fig. 1 | Nanowire device geometry.** Schematic diagram of a horizontal InAs (**a**) multi-faceted NW and (**b**) single-faceted NW grown on the top of a GaAs nanoridge. **c**, **d** HAADF-STEM micrographs of the NW cross sections, corresponding to (**a**) and (**b**), respectively, showing InAs NW (bright grey) and GaAs nanoridge (dark grey). SEM images of the fabricated InAs NW receivers: **e** 1NW receiver; **f** 5NW receiver; (**g**) 20NW receiver.

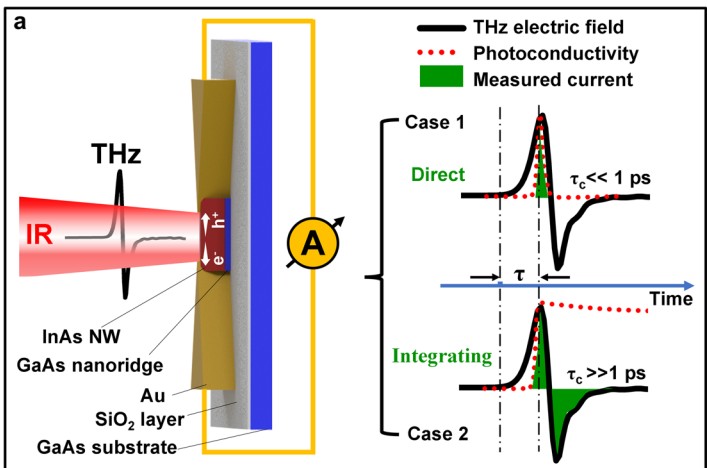
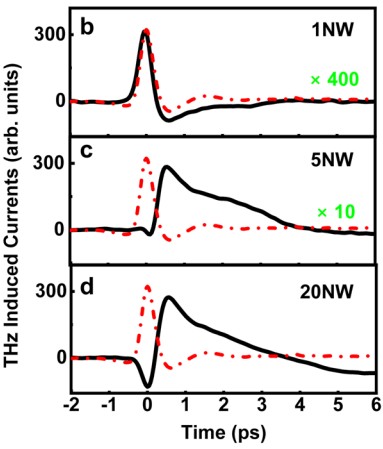

**Fig. 2 | Direct vs. integrating sampling. a** Schematic diagram of a photoconductive NW THz receiver in operation, whose THz response for a detection material with (case 1) short carrier lifetime (<<1 ps) and with (case 2) long carrier lifetime (>>1 ps) illustrated on the right. **b**–**d** Raw data of THz responses obtained from a multi-faceted 1NW receiver, a single-faceted 5NW receiver and a single-faceted 20NW receiver, respectively. All above receivers have an identical antenna gap size of 1 μm and were optically gated with 1.5-μm laser pulses of 1.66 mJ/cm² fluence. The red dash-dotted line represents raw data of THz response obtained from an ion-implanted InP bulk receiver (as reference), which was optically gated with 800-nm laser pulses of 1.5 mJ/cm² fluence.

where, the measured current is directly proportional to the strength of the incident THz electric field at that point.

Case 2 (integrating sampling): the detection material has a long photoconductivity lifetime (>>1 ps), the conductivity $\sigma(\tau)$ of the detection material can be approximated as a unit step function and therefore, the measured current is proportional to the integral of the electric field of the THz pulse, expressed by

$$I(\tau) \propto \int_{\tau}^{\infty} E_{(THz)}(t)dt \qquad (3)$$

where, the THz electric field $E_{(THz)}(t)$ can be recovered from the $I(\tau)$ data by differentiation with respect to time.

When the charge-carrier lifetime of the detection material is comparable to the THz pulse duration (here we consider the range of 1 ps < $\tau$ < 100 ps), all the above approximations fail. In this scenario, $E_{(THz)}(t)$ can only be extracted by numerical deconvolution (see ref. 66).

Figure 2b–d shows the measured THz-induced current transients under 1.5-μm optical gating for 1NW, 5NW and 20NW receivers, respectively. Moving from 1NW to 20NW receivers, the current signal increases about 400 times, leading to an associated increase in signal-to-noise ratio (SNR)[62,63]. An increase is generally expected as the current increases with the total detection material volume. Interestingly, the THz-induced currents exhibit significantly different profiles. The 1NW receiver exhibits a very short response, similar to the original THz pulse. This is characteristic of the direct sampling mode. The multiple NW devices exhibit a broader temporal response extending to 6 ps or longer, coherent with the integrating sampling mode.

The sampling mode was confirmed by measuring the THz pulse with a Fe⁺-implanted bulk InP photoconductive THz receiver. We fabricated this reference receiver using the same bowtie antenna structure as the NW receivers, so that the (weak) spectral filtering of the THz radiation by the gold antenna would be similar across all devices. We measured the photoconductive response, $\sigma(t)$, of the Fe⁺-implanted bulk InP using optical-pump-THz-probe spectroscopy[67], and found the photoconductivity lifetime to be 400 fs[66], which means the reference InP receiver operates in direct sampling mode and so the raw current data are a good representation of the true THz electric field, without the need of further data processing (see more details in Supplementary Information, Figs. S4 and S5). By comparing Fig. 2b–d with the InP reference (see red dash-dotted line in Fig. 2), the shape of the raw current data from the 1NW receiver is indistinguishable from that of the InP reference, indicating that multi-faceted 1NW receivers operate in direct sampling mode. In contrast, single-faceted 5NW and 20NW receivers are found to operate in integrating sampling mode, with the true form of the THz transient recovered by differentiation with respect to time (see more discussions in Supplementary Information, Figs. S4 and S5). We also explored the THz response of both the 1NW and 5NW receivers under varying gate wavelengths, as shown in Fig. 3a. An integrating sampling response was obtained for optical gating wavelengths ranging from 1.2 to 1.55 μm for the 5NW THz receiver, and a direct sampling response was obtained for the 1NW THz receiver (More details are provided in Supplementary Information, Fig. S6).

These results remarkably indicate that 1NW receivers have extremely short photoconductivity lifetime with respect to 5NW and 20NW receivers even though all the InAs NWs are grown under nominally identical growth conditions. To explain this difference, we consider three main factors that can affect the lifetime to this extent: surface-to-volume ratio, defect density and surface recombination velocity of different facets. The cross-sectional HAADF-STEM images shown in Fig. 1b, d indicate that there is not a major difference in the InAs thickness (cross-sectional area varies from 14,400 nm² to 15,200 nm²), nor surface-to-volume ratios (ranging from 0.023 and 0.036 nm⁻¹). HR-STEM (reported in Supplementary Information, Figs. S7 and S8) does not evidence significant defect-related differences among the devices. On the contrary, there is a stark correlation between the sampling mode and the surface morphology of the NWs, indicating that wires with high-index facets exhibit a shorter photoconductivity lifetime due to higher surface recombination velocity. Hence, we identify surface recombination on different crystal surfaces as the key factor which determines the different photoconductivity lifetime of the multi-faceted and single-faceted horizontal InAs NWs.

We further investigate the role of surface recombination on the performance of the InAs NW THz receivers. For this, we perform excitation fluence-dependence experiments. Figure 3b shows the variation of the response obtained from a 5NW receiver optically gated by a 1.2 μm-wavelength laser with pulse fluences of 0.05, 0.98 and 2.7 mJ/cm². The temporal profile of the transient current response moves from a short to a broader response. We interpret this as a transition from direct sampling mode to integrating. This indicates

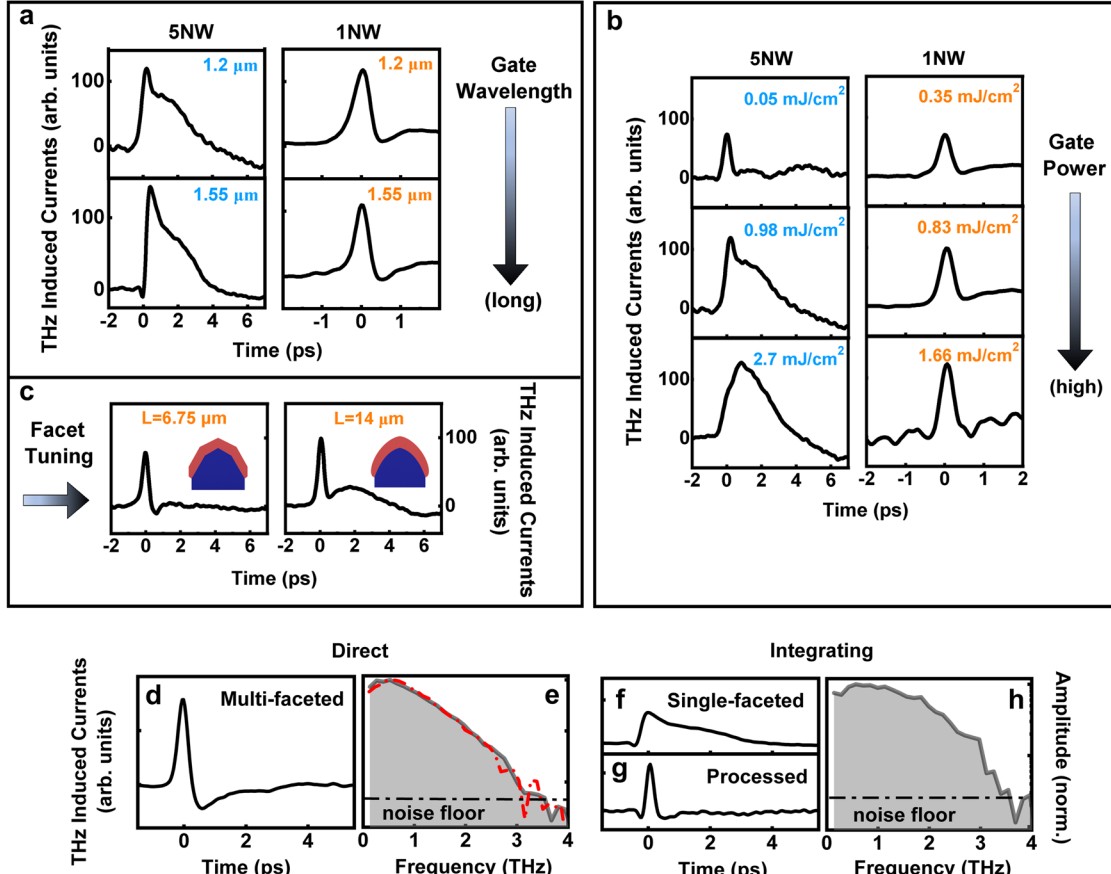

**Fig. 3 | THz characterisation. a** Comparison of raw data of THz responses obtained from a 5NW and a 1NW receiver under varying excitation wavelengths of 0.98 mJ/cm² fluence, respectively. **b** Comparison of raw data of THz responses obtained from a 5NW (left) and a 1NW (right) receiver under varying excitation fluences. The 5NW is optically gated by a 1.2 μm-wavelength laser and the 1NW receiver is optically gated by a 1.55 μm-wavelength laser. **c** Comparison of raw data of THz responses obtained from 1NW receivers made of a 6.75-μm-long NW (identified as multi-faceted) and a 14-μm-long NW (identified as undeterminable-faceted). **d** Raw data of THz response obtained from a multi-faceted 1NW receiver and its corresponding THz spectral response as shown in (**e**). **f** Raw data of THz response from a single-faceted 5NW THz receiver and its recovered THz transient as shown in (**g**) by differentiation of data in (**f**), which was further converted into THz spectral response as shown in (**h**). Both receivers were optically gated by a 1.5-μm-wavelength laser under fluence of 0.98 mJ/cm².

that the photoconductivity lifetime of the single-faceted NWs increases with the gate pulse fluence. This result is consistent with saturable-trap-mediated Shockley–Read–Hall (SRH) recombination dominating charge-carrier recombination[42,67]. The sensitivity to excitation fluence is independent of the excitation wavelength (See Supplementary Information, Fig. S9).

Figure 3b also displays the response of the 1NW receiver over the fluence range from 0.35 to 1.66 mJ/cm², optically gated by a 1.55 μm-wavelength laser. Here, the temporal response of the raw current barely changes under varying excitation fluence. The single NW device exhibits direct sampling mode for all gating fluences investigated. The very short charge carrier lifetime combined with no observable saturation over our fluence range indicates a very high density of SRH recombination centres in multi-faceted NWs. This high trap density could be related to an increased number of dangling bonds related to the presence of {311} facets in these NWs. The critical role of the InAs faceting on the photoconductivity lifetime is further confirmed by the observation of a transition from direct to convolved sampling mode observed on some 1NW receivers as shown in Fig. 3c, characterised by micro facets whose orientations are undeterminable (See Supplementary Information, Fig. S11). We correlate the appearance of this morphology to the increase of the nominal length from 6.75 μm to 14 μm in nanowire growth, which effectively modifies the growth rate of the underlying GaAs nanoridge template and therefore tuned the facet types for InAs nanowires.

We now turn to examine the responses of the 1NW and 5NW receivers to demonstrate it is possible to recover the THz pulse. Figure 3d, f reports the raw THz-induced current data from 1NW and 5NW receivers, respectively. Figure 3g displays the response of single-faceted 5NW receiver processed according to Eq. (3). The similarity in the recovered THz temporal waveform profiles from both receivers, (see Fig. 3d, g), confirms our assumption of the sampling modes and the associated photoconductivity lifetime of the NWs.

Finally, the recovered THz pulse data was then Fourier transformed to obtain the THz spectral response for the THz receivers with multi-faceted 1NW and single-faceted 5NW receivers. This is presented in Fig. 3e and 3h, respectively. Both spectral responses show a broadband response approaching 3 THz (defined as the cut-off frequency at the noise floor of its spectral response), comparable with bulk THz receivers in literature[68]. The multi-faceted 1NW receiver presents a smooth spectrum in the frequency domain, owing to its direct sampling nature. This avoids the inclusion of additional uncertainties associated with the processes of numerical differentiation or deconvolution. In comparison, the spectrum of the single-faceted 5NW receiver, which requires additional data processing (see Fig. 3g), gives rise to artefacts (this is inherent to the digitisation and differentiation process), seen as waviness in both the time and frequency domain plots (see Fig. 3g, h). Moreover, despite showing a hundred times lower measured signal strength, the multi-faceted 1NW receiver shows a comparable performance to a direct

sampling InP bulk receiver in terms of detection bandwidth (see the red dash-dotted line in Fig. 3e) under the operation conditions used, indicating its feasibility in practical applications. A detailed comparison between different types of NW receivers can be found in Supplementary Information, Table S1.

Over the past few decades many studies have searched for an ideal semiconductor with both ultrashort photocarrier lifetime and high photocarrier mobility for low-noise photoconductive THz detection via direct sampling. For this, low-temperature-growth[69] and ion-implantation techniques[66,70] have been developed. The former has a reproducibility issue, and the latter involves multiple rounds of ion-implantation and annealing steps, which makes fabrication challenging and expensive. The intrinsic high surface-to-volume ratio of semiconductor NWs gives rise to a shorter charge-carrier lifetime compared to their bulk counterparts while maintaining a bulk-like high charge-carrier mobility via optimised growth conditions. One could in principle argue the use of top-down etching of InAs NWs for the similar purpose. However, this approach does not allow for facet engineering[71] and requires special processing to avoid damage to the NW surface from the etching. Thus, the proposed bottom-up NWs are excellent material candidates to meet the demand for direct sampling of THz radiation. This work, for the first time, verified this idea via the development of facet-engineered horizontal InAs NWs, which should inspire new THz sensing and imaging technologies.

This work demonstrates InAs-NW time-domain THz receivers, which have not been reported to date. We tested the NW receivers with optical gate lasers with wavelengths ranging from 1.2 to 1.55 μm, indicating the suitability of our receivers for integration with ultrafast fibre lasers at telecom wavelengths at room temperature. The receivers are based on scalable selective-area-grown horizontal NWs with low defect density. We demonstrate how these NWs can be engineered to enable either direct sampling or integrating sampling modes for optimal THz sensing. The results strongly suggest that the faceting of the NWs is the key factor governing the sampling mode. At the same time, both direct and integrating sampling receivers exhibit a detection bandwidth of up to 3 THz. As opposed to previous NW THz receivers, the lateral patterning technique utilised in this work is highly scalable and is suitable for the fabrication of large-area sensor arrays. Thus, this work paves the way to develop real-time broadband THz cameras made of nano-receiver focal plane arrays for high-speed, high-resolution THz imaging applications. Significantly, in contrast to semiconducting films, NWs are intrinsically polarisation-sensitive to THz radiation[39] and naturally form low dark current, electrically isolated THz receiver elements[41]. These features make our scalable InAs NW technology promising for the development of polarisation-sensitive THz cameras thereby enabling future THz polarisation imaging applications.

## Methods
### Growth
Intrinsic, semi-insulating GaAs (100) substrates were prepared by first patterning and depositing W alignment markers using e-beam lithography, sputtering and lift-off. Next, 27 nm of $SiO_2$ was deposited by plasma-enhanced chemical vapour deposition. This was followed by e-beam lithography using ZEP resist and low-temperature development to achieve low-line edge roughness. Subsequent reactive ion etching with fluorine chemistry was used to etch the $SiO_2$ down to the GaAs surface in the patterned trenches, and a final wet etch in a dilute buffered HF solution was performed to remove any remaining oxide in the trenches and to smooth the surface roughness of the oxide mask. The resulting substrate featured a $SiO_2$ mask about 24 nm in thickness with openings ranging from 40 to 200 nm in nominal width and lengths from 6 to 40 μm, depending on the pattern.

Growth of the nanostructures was performed using molecular beam epitaxy (MBE) in a DCA P600 growth chamber using Ga and In effusion cells and an As valved cracker sublimation cell. The sample holders were first heated to 400 °C for 2 hours in a separate degas module to remove any residual organic contamination from the surface. Once in the growth module, annealing was performed at 630 °C under $As_4$ flux with a beam equivalent pressure (BEP) of $6 \times 10^{-6}$ Torr to remove native oxides. First, GaAs nanoridges are grown at 630 °C with a 2D-equivalent growth rate of 0.3 Å/s and a V/III BEP ratio of 80. The GaAs growth lasts 55 minutes, at which point the Ga flux is stopped and the substrate holder is cooled to 520 °C under $As_4$ flux. Once at the appropriate temperature, the In flux is started and the InAs growth proceeds with a 2D-equivalent growth rate of 0.35 Å/s and a V/III ratio of 10. After 19 min of growth, the In flux is stopped and the sample is cooled under $As_4$ flux before removal from the growth chamber for device fabrication.

### Fabrication details
To fabricate devices, chips were spin-coated with an MMA/PMMA bilayer resist and patterned using e-beam lithography. After development and a short descum treatment in oxygen plasma, the metal bowtie antenna electrodes were deposited by room-temperature sputtering. The metal stack includes a 10 nm layer of chromium for adhesion and 140 nm of gold. After lift-off in acetone, the devices were rinsed in isopropyl alcohol and ready for characterisation.

### TEM characterisation
TEM lamellae were created by focused ion beam using a Zeiss NVision 40 CrossBeam. STEM-HAADF imaging was performed using a Thermo Fisher Scientific Talos F200S with 200 kV acceleration voltage and 1 nA probe current. High-resolution STEM HAADF imaging was performed in an aberration-corrected FEI Titan Themis with 300 kV acceleration voltage and 100 pA probe current. Geometric phase analysis (GPA) was realised on the HR-STEM images using in-house GMS scripts.

### THz time-domain spectroscopy system
An amplified laser system (Spectra Physics, MaiTai - Empower - Spitfire) with an average power of 4 W, central wavelength of 800 nm, pulse duration of 35 fs and repetition rate of 5 kHz was used. The laser beam was split into two beam paths: one was guided to a spintronic THz emitter (emission via the inverse spin-Hall effect) to generate a single-cycle linearly-polarised THz pulse in the system (See ref. 25 for details); the other, going through a TOPAS-OPA to allow wavelength tuning from 290 nm to 2.6 μm, was guided to the receiver to generate photocarriers. A schematic of the THz time-domain spectroscopy setup is presented in Fig. S3. The THz pulse was directly detected by the NW receivers. All measurements were repeated at least 3 times, from which the uncertainty was determined by the standard deviation.

In our experiment, the maximum average output power from TOPAS-OPA exciting on the NW receivers is 38 mW, 42 mW, 31 mW, 18.5 mW and 15.5 mW when operating at the wavelength of 1.2 μm, 1.3 μm, 1.4 μm, 1.5 μm and 1.55 μm, respectively. Under different laser excitation wavelengths and powers, the corresponding spot sizes vary a bit. For example, when optically gating at 1.55 μm, the pulse fluence of 0.35 mJ/cm$^2$ corresponds to the excitation power of 2.3 mW on average with a laser spot size in diameter of 410.8 μm ($1/e^2$ width). Therefore, throughout the manuscript, we use laser fluence to describe our experimental condition, which takes into account the excitation wavelength, power, spot size and pulse repetition rate. This helps to compare and evaluate different experimental conditions.

## Data availability
The data that support the findings of this study are available from the corresponding author upon request.

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

## Acknowledgements

We thank the EPSRC (UK) for funding this project through grants EP/W018489/1 and EP/T025077/1 (M.B.J.). The authors from EPFL acknowledge funding from the Swiss National Science Foundation, including from the NCCR QSIT (A.F.i.M.). V.P. gratefully acknowledges the funding from Piaget. Authors also thank EPFL facilities for nanofabrication and electron microscopy characterisation, respectively CMi and CIME.

## Author contributions

K.P. conducted the experimental design, THz measurements, data analysis and FDTD simulations. N.P.M. grew the NWs, fabricated NW devices, and performed electron microscopy and analysis. C.Q.X., F.W. and T.S. assisted with THz system development and measurements. D.D. assisted with NW growth and device fabrication. V.B. performed HR-STEM, GPA, and defect analysis. V.P. performed data analysis. The manuscript was drafted by K.P. and N.P.M. under the supervision of VP, MBJ and AFiM. The project concept was devised by K.P., M.B.J. and A.F.i.M.

## Competing interests

The authors declare no competing interests.
