## [Peer Review File · Nature Communications]

Direct and Integrating Sampling in Terahertz Receivers from Wafer-Scalable InAs NanowiresREVIEWER COMMENTS

Reviewer #1 (Remarks to the Author):

THz radiation detectors made of narrow bandgap InAs semiconductor are described. In such a detector, high sensitivity is ensured by a high electron mobility of the crystal, high dark resistance is ensured by using horizontally grown nanowires (NW), and required short carrier lifetimes – by the surface recombination in planes of low symmetry on NW surfaces. The relationship between the presence of multi-faceted NW and short lifetimes is shown.

However, the mechanism of the formation of such multi-faceted NWs is not sufficiently described in the article - it remains unclear whether their emergence can be controlled technologically, or whether it is simply a random phenomenon. In the given implementation example, multi-faceted NW is single, while single-faceted NWs are multiple. Cross-section of the GaAs nanoridge, on which multi-faceted NW grows, is non-standard and rather irregular; the reason for its formation is also unclear.

It is worth paying more attention to these essential details of the technological processes, instead directing the reader to References or other parts of the article, which also do not contain such information. Especially since such NWs with short carrier lifetimes are critically important in the development of THz imaging devices, which are indicated in several places of the article as potentially most important application of the obtained results.

Therefore, I recommend adding the above explanations to the manuscript before accepting it for publication. Also the rather trivial descriptions of the THz induced photocurrent could be shortened.

Reviewer #2 (Remarks to the Author):

Peng et al. demonstrate photoconductive THz receivers based on InAs nanowires operating at telecom wavelengths. The main significance of the work is the scalability by using MBE grown horizontal InAs nanowires selectively defined by ebeam lithography. It is also shown that the facet type of the grown InAs nanowires depends on the density and/or length of the prior-grown GaAs nanoridges, resulting in remarkable different carrier lifetime and THz detection (direct vs. integrating samplings).

I want to provide following comments/questions that are relevant to the claimed scalability:

1. Are the GaAs nanoridges necessary for growing the InAs nanowires? In other words, can InAs nanowires be defined by the SiO₂ mask trenches and grown directly on the exposed GaAs surface? If yes, would we end up with (100) InAs nanowires and expect different carrier lifetime and mobility?

2. Since the patterned growth of InAs nanowires already involves ebeam lithography, one may imagine the fabrication of InAs nanowires via direct etching of InAs film. Can the author comment on etching vs. growth?

3. In order to illustrate the significance of scalable nanowire-based THz receiver, it is important for readers to understand the benefits of using nanowires as compared to other more matured material platforms, e.g., Er-doped GaAs or InGaAs via MBE growth. The latter is well suitable for scalable THz receivers.

4. An amplified laser is used in this work. How much is the power from the TOPAS? Would typical telecom femtosecond (fiber) lasers be applicable, particularly when unfocused for large area

illumination or splitting into multiple focal spots for THz camera applications?

Reviewer #3 (Remarks to the Author):

The authors report on successful realization of photoswitches utilizing laterally grown InAs-based nanowires. This work advances the field of nanowire-based devices in several aspects: firstly, the specificity of material allows for employing well-developed fibre-based lasers, secondly, the presented method of manufacturing laterally-oriented nanowires allow for fabrication of arrays scalable to the large number of elements, thus showing the potential for developments of novel, ultra-fast focal-plane array imagers and etc.,

The technological part of the work is well presented, concise and contains enough details in the methods/supplementary parts for to be traced.

The experimental data supports a remarkable finding - that the fabricated InAs (mostly single) nanowire structures possess short photoconductivity lifetime, which is detrimental for their further utilization as a photoswitch device. What is a bit harder to trace from the presented investigations - is the origin of increased lifetime in multi-nanowire devices. The authors perform a thorough study of thickness and state that no major difference as well as no significant defect-related differences between devices can be observed. Yet, the explanation for the difference in observed decay of response between a single and multiple nanowire photoswitches is attributed to the saturation of trap-mediated recombination. There might be the possibility, that the broadening of the response might originate from the differences in travel distances between the individual current channels in multi-wire devices, yet this situation does not seem to be simulated. It would be also interesting/important to see the measured differences in signal-to-noise ratios for different kinds of devices. Although the photocurrent for multi-nanowire devices is obviously larger, the lower conductivity results in larger current noise, thus the necessity for multi-nanowire devices isn't self-explanatory. A comprehensive comparison between conductivities, photocurrents and current noise would be advantageous. Additional support of the concept might come from electrodynamic simulations of THz power absorbed by a single nanowire and their ensemble. In the current presentation, it is hard to deduce in which case the spectral data shows more dynamics, but there is an impression, that the dynamical range of the sensitivity for a single nanowire device is not underperforming.

I would like to agree with the conclusion of the authors that the proposed novel devices under the similar excitation conditions might become competitive to ion-implanted bulk InP receivers, however, in order to support the claim that the proposed technological approach can become a break-through technology, a comparison with state-of-the-art photoswitches outside of nanowire research subfield is mandatory. That might also require revisiting the radiation coupling scheme. To my knowledge, a front-side coupling of THz radiation to a broadband antenna as it was employed by the authors is inferior to a well-aligned substrate lens configuration which is routinely implemented in commercial photoconductive THz emitters and receivers.

REVIEWER COMMENTS**Reviewer #1** (Remarks to the Author):

THz radiation detectors made of narrow bandgap InAs semiconductor are described. In such a detector, high sensitivity is ensured by a high electron mobility of the crystal, high dark resistance is ensured by using horizontally grown nanowires (NW), and required short carrier lifetimes – by the surface recombination in planes of low symmetry on NW surfaces. The relationship between the presence of multi-faceted NW and short lifetimes is shown.

However, the mechanism of the formation of such multi-faceted NWs is not sufficiently described in the article - it remains unclear whether their emergence can be controlled technologically, or whether it is simply a random phenomenon.

We agree with the referee that this is an important aspect of the manuscript. We confirm that the formation of multi-faceted NWs can be controlled as it is a matter of stage in the growth. It can be controlled with time or, for a given growth time, by the width and pitch of the trenches. An example is illustrated in the SEM images below. For a given pitch and growth time, the GaAs nanowires belong to different stages of growth and different faceting. More details are found in Refs 48 and 49.

In the given implementation example, multi-faceted NW is single, while single-faceted NWs are multiple. Cross-section of the GaAs nanoridge, on which multi-faceted NW grows, is non-standard and rather irregular; the reason for its formation is also unclear. It is worth paying more attention to these essential details of the technological processes, instead directing the reader to References or other parts of the article, which also do not contain such information. Especially since such NWs with short carrier lifetimes are critically important in the development of THz imaging devices, which are indicated in several places of the article as potentially most important application of the obtained results. Therefore, I recommend adding the above explanations to the manuscript before accepting it for publication.

The multi-faceted nature of the nanowires is a consequence of the natural crystal growth progression under the chosen growth conditions. Nanowire formation begins with the nucleation of three-dimensional islands which are defined by {311} facets, followed by coalescence and further development of the nanoridge.

The cross-section TEM here below from Ref. 48 show the facet formation at two different stages of growth from a mixture of {311} and {111} families to predominantly {111}.

Upon deposition of InAs on the GaAs ridge there is interdiffusion that roughens the interface, especially for high index facets. This has been demonstrated in various works, for example in M. Friedl et al, Nano Lett 18, 2666 (2018).

Given the importance of this aspect, we have now added an explanation in the main text of the manuscript as well as in **Supplementary Information**.

*Proposed changes, addition of the following text page 5:

“Whether a NW is multi-faceted or single-faceted depends on the stage of development of the underlying GaAs nanoridge at the moment that the growth switches from GaAs to InAs. If the GaAs nanoridge is underdeveloped, a multi-faceted NW will arise. If the GaAs nanoridge is fully developed, a single-faceted NW will result. The stage of development of the GaAs nanoridge is a function of its local growth rate, which in turn is dependent on the presence and proximity of neighbouring nanoridges, and the growth time, as described in ref. [48]. This allows for precise control over whether a NW is single- or multi-faceted.”

and

“By adjusting the growth time, it would be equally possible to create parallel arrays of multi-faceted NWs or isolated single-faceted NWs, though this is not achievable on a single growth substrate.”

Changes in the Supplementary Information, page 10:

“From the STEM images we see that whether an InAs NW is single- or multi-faceted is determined by the morphology of the underlying GaAs nanoridge. In the case of multi-faceted NWs, the GaAs nanoridge is underdeveloped, exhibiting roughness and morphological irregularity. For a fixed growth time, the stage of development of the GaAs nanoridge is a function of its local growth rate, which in turn is dependent on the presence and proximity of neighbouring nanoridges, as described in ref. [48] of the main text. Growth time can also be adjusted to independently control the faceting of the NW, regardless of whether it is grown in isolation or in a parallel array.”

Also the rather trivial descriptions of the THz induced photocurrent could be shortened.

We thank the referee’s suggestion. We aimed to make the manuscript as clear as possible to readers without a knowledge of THz devices which is why we included a self-contained description of how our devices work, but agree we could have been more concise. We have made our descriptions more concise. Please see the changes to pages 6, 7 and 8 of the manuscript.

Reviewer #2 (Remarks to the Author):

Peng et al. demonstrate photoconductive THz receivers based on InAs nanowires operating at telecom wavelengths. The main significance of the work is the scalability by using MBE grown horizontal InAs nanowires selectively defined by ebeam lithography. It is also shown that the facet type of the grown InAs nanowires depends on the density and/or length of the prior-grown GaAs nanoridges, resulting in remarkable different carrier lifetime and THz detection (direct vs. integrating samplings).

I want to provide following comments/questions that are relevant to the claimed scalability:

1. Are the GaAs nanoridges necessary for growing the InAs nanowires? In other words, can InAs nanowires be defined by the SiO₂ mask trenches and grown directly on the exposed GaAs surface? If yes, would we end up with (100) InAs nanowires and expect different carrier lifetime and mobility?

This is indeed a fair question. Previous works have shown that the electronic properties of InAs nanowires obtained directly on a patterned substrate are directly determined by the contamination at the interface (inherent in any commercial substrate due to the polishing process)¹. The introduction of the GaAs ridge acts as a buffer layer, increasing reproducibility and the properties of the InAs. In addition, it is only by controlling the GaAs nanoridge that we control the final faceting of the InAs NW in a way that it is decoupled from the NW thickness.

To clarify this aspect for the readers, we have now added the following in the main text of the manuscript

*Proposed changes, addition to the text on page 4:

“Our InAs NWs are grown horizontally on GaAs nanoridges via selective-area molecular beam epitaxy (MBE), an approach which physically separates the active NW component from impurities present on the substrate surface due to previous process steps [51].”

2. Since the patterned growth of InAs nanowires already involves ebeam lithography, one may imagine the fabrication of InAs nanowires via direct etching of InAs film. Can the author comment on etching vs. growth?

We thank the referee for the interesting question. Producing InAs nanowires directly by etching would have the following disadvantages:

- (i) We would need to start with a InAs substrate or alike, requiring special engineering to separate the nanowires electronically and optically from the substrate. InAs substrates are also significantly more expensive than GaAs due to the scarcity of In.
- (ii) According to literature on top-down etching of nanowires, there is today no known possibility to engineer faceting.²

¹ Filip Krizek PhD thesis, U. Copenhagen https://nbi.ku.dk/english/theses/phd-theses/phd_theses_2018/filip_krizek/Filip_Krizek.pdf

² P.C. McIntyre, A. Fontcuberta i Morral, Semiconductor nanowires: to grow or not to grow?, Materials Today Nano, 9, 100058 (2020)

We have now added these points in the introduction of the manuscript plus inserted a new ref [71].

*Changes page 11, addition of the following text:

“One could in principle argue the use of top-down etching of InAs nanowires for the similar purpose. While current state does not allow for facet engineering [71], it would also be challenging to isolate those nanowires optically and electrically from the rest of the substrate. Thus, the proposed bottom-up nanowires are excellent material candidates to meet the demand for ‘direct-sampling’ of THz radiation.”

3. In order to illustrate the significance of scalable nanowire-based THz receiver, it is important for readers to understand the benefits of using nanowires as compared to other more matured material platforms, e.g., Er-doped GaAs or InGaAs via MBE growth. The latter is well suitable for scalable THz receivers.

The referee makes a good point here. One significant benefit of using nanowires (rather than bulk materials) for THz detection is to allow fast and precise polarisation-sensitive THz measurements (see ref [41] in main text), which result in higher accuracy in THz spectroscopy and higher image contrast and higher spatial resolution in THz imaging.

We mentioned this briefly in the **Introduction**, (see page 3: the last sentence of the **Introduction**), but agree with the referee that it is important to emphasize this point and contrast with thin films. We have now cited ref [41] in this sentence.

*Changes page 3, citation of ref [41] in the following sentence:

“The fabrication scalability and properties tunability of the horizontal InAs NWs also promise room for further functionalization via pattern engineering and polarisation-sensitive detection [41].”

and

*Changes page 2, following sentences added:

“Furthermore, the geometry of NWs can further aid absorption [39, 40], allowing constraints to be relaxed on laser wavelengths suitable for achieving switching photoconductive devices. As opposed to conventional bulk and thin-film based photoconductive THz receivers, NWs offer excellent polarisation sensitivity to THz radiation [39], combined with negligible electrical cross-talk between detector elements and lower dark current, making them ideal for THz receiver arrays or multi-channel receivers [41].”

We also agree that we should highlight the capability of polarization sensing by NW THz receivers and thus we added some comments in the **Conclusion**.

*Changes page 11, addition of the following text:

“Significantly, in contrast to semiconducting films, NWs are intrinsically polarisation-sensitive to THz radiation [39] and naturally form low dark current, electrically isolated THz receiver elements [41]. These features make our scalable InAs NW technology promising for the development of polarisation-sensitive THz cameras thereby enabling future THz polarisation imaging applications.”

4. An amplified laser is used in this work. How much is the power from the TOPAS? Would typical telecom

femtosecond (fiber) lasers be applicable, particularly when unfocused for large area illumination or splitting into multiple focal spots for THz camera applications?

We thank the referee for this good question. Indeed telecoms femtosecond fibre lasers would be better sources for this particular receiver, however we unfortunately did not have access to one for this study. Hence used an optical parametric amplifier (TOPAS OPA) to simulate the pulses for a wide range of possible fibre lasers, at the cost of lower signal to noise ration. Actually, all photoconductive THz sensors are better suited to higher repetition rate laser (such as femtosecond fibre lasers, or laser oscillators) rather than more noisy, lower repetition rate amplified lasers or even noisier OPAs. In our experiment, the maximum average output power from TOPAS exciting on the NW receivers is 38 mW, 42mW, 31mW, 18.5 mW and 15.5 mW when operating at the wavelength of 1.2 μm , 1.3 μm , 1.4 μm , 1.5 μm and 1.55 μm , respectively. We have now added the information in the **Method**.

*Changes pages 12 and 13, addition of the following text:

“In our experiment, the maximum average output power from TOPAS-OPA exciting on the NW receivers is 38 mW, 42mW, 31mW, 18.5 mW and 15.5 mW when operating at the wavelength of 1.2 μm , 1.3 μm , 1.4 μm , 1.5 μm and 1.55 μm , respectively. Under different laser excitation wavelengths and powers, the corresponding spot sizes varies a bit. For example, when optically gating at 1.55 μm , the pulse fluence of 0.35 mJ/cm^2 corresponds to the excitation power of 2.3 mW in average with a laser spot size of 410.8 μm in diameter ($1/e^2$ width). Therefore, throughout the manuscript we use laser fluence to describe our experimental condition, which has been taken into account of the excitation wavelength, power, spot size and pulse repetition rate. This helps to compare and evaluate different experimental conditions.’

When optically gating at 1.55 μm , it was straightforward to achieve good THz signals from a 1NW receiver under a pulse fluence of 0.35 mJ/cm^2 using the amplified laser (at 5kHz repetition rate), which corresponds to the excitation power of 2.3 mW with a laser spot size of 410.8 μm in diameter ($1/e^2$ width). Typical 1.55- μm fibre fs lasers used in THz spectrometer use ~ 10 mW to gate commercial THz receivers typically with 100fs pulses at 40-80MHz repetition rate. For photoconductive THz receivers it is the average power that matters in terms of total signal acquired and the higher repetition rate of fibre lasers generally improves the signal-to-noise ratio of measurements. In addition, by means such as increasing the power of fiber laser from 2.3 mW to 23 mW, reducing the laser spot size to 10 times smaller and adding more numbers of NWs in detection channel, it is practical to reach a reasonable excitation condition with femtosecond fibre lasers for NW receiver/receiver-array operation. Fibre lasers with averages power exceeding 100mW are also readily available for the gating of larger arrays.

Reviewer #3 (Remarks to the Author):

The authors report on successful realization of photoswitches utilizing laterally grown InAs-based nanowires. This work advances the field of nanowire-based devices in several aspects: firstly, the specificity of material allows for employing well-developed fibre-based lasers, secondly, the presented method of manufacturing laterally-oriented nanowires allow for fabrication of arrays scalable to the large number of elements, thus showing the potential for developments of novel, ultra-fast focal-plane array imagers and etc., The technological part of the work is well presented, concise and

contains enough details in the methods/supplementary parts for to be traced. The experimental data supports a remarkable finding - that the fabricated InAs (mostly single) nanowire structures possess short photoconductivity lifetime, which is detrimental for their further utilization as a photoswitch device.

What is a bit harder to trace from the presented investigations - is the origin of increased lifetime in multi-nanowire devices. The authors perform a thorough study of thickness and state that no major difference as well as no significant defect-related differences between devices can be observed. Yet, the explanation for the difference in observed decay of response between a single and multiple nanowire photoswitches is attributed to the saturation of trap-mediated recombination.

We considered the thickness and defect density variations as possible causes for the increase of lifetime but evidence indicates that they are quite small in magnitude. On the contrary, there is a straightforward correlation with the faceting of the NWs. III-V materials are known to exhibit different densities of Shockley-Reed-Hall recombination centers on different facets. This is particularly important for InAs, which exhibits Fermi level pinning and a corresponding accumulation of electrons at its free surfaces. We therefore conclude that lifetime is mostly determined by the surface states at different facets.

We have now clarified this point in the main manuscript:

*Changes page 8, addition of the following text:

“On the contrary, there is a stark correlation between the sampling mode and the surface morphology of the NWs, indicating that wires with high-index facets exhibit a shorter photoconductivity lifetime due to higher surface recombination velocity.”

There might be the possibility, that the broadening of the response might originate from the differences in travel distances between the individual current channels in multi-wire devices, yet this situation does not seem to be simulated.

Although we have not simulated this possibility, we have considered it and eliminated it as the cause of the difference in lifetime. We have done this by preparing single NW devices with varying lengths (spacing between the electrodes) ranging from 0.75 to 8 μm . We present this data in Figure S10 and comment on it in the text of the manuscript. We find that even for electrode spacings up to 8 μm , we observe short lifetimes with direct type detection behavior. Thus, the travel distance of the carriers cannot explain the long carrier lifetimes we observe in the multi-NW devices, because in such a case we would also observe increasing carrier lifetime with increasing electrode spacing in the single NW devices.

It would be also interesting/important to see the measured differences in signal-to-noise ratios for different kinds of devices. Although the photocurrent for multi-nanowire devices is obviously larger, the lower conductivity results in larger current noise, thus the necessity for multi-nanowire devices isn't self-explanatory. A comprehensive comparison between conductivities, photocurrents and current noise would be advantageous. Additional support of the concept might come from electrodynamic simulations of THz power absorbed by a single nanowire and their ensemble. In the current presentation, it is hard to deduce in which case the spectral data shows more dynamics, but there is an impression, that the dynamical range of the sensitivity for a single nanowire device is not underperforming.

This is an excellent point. We have now added Table S1 in **Supplementary Information** to compare different NW receivers studied in this work regarding their THz detection performance under the same excitation condition.

*Changes page 10, addition of the following text:

“A detailed comparison between different types of NW receivers can be found in Supplementary Information, Table S1.”

And changes to page 7 in the **Supplementary Information**, addition of the following text:

“After confirming the sampling mode of each type of receivers, we processed their raw THz response data accordingly to evaluate the performance of the NW receivers, as shown in Table S1.

Table S1. Comparison of detection performance between 1NW, 5NW and 20NW receivers. (For signal-to-noise ratio, the signal is defined as the peak-to-peak current over one time-domain scan, and the noise is the standard deviation of the difference of two consecutive scans with identical parameters. Dynamic range is defined as the ratio of the peak-to-peak current over one time-domain scan to the standard deviation of the noise current in the absence of THz over the same scan.) All NW receivers have an identical antenna gap size of 1 μm and were optically gated with 1.5- μm laser pulses of 1.66 mJ/cm^2 fluence.

	Detector Type		
	Photoconductive Antenna (bow tie)		
Performance	1NW	5 NW	20 NW
NW Facet Type	Single-faceted	Multi-faceted	Multi-faceted
Sampling Modes	Direct	Integrating	Integrating
Spectral Response Bandwidth (THz)	3.0	3.0	3.0
Signal-to-noise Ratio	6.0	21.8	27.8
Dynamic Range	93.9	94.1	140

*Signal-to-noise ratio determines the accuracy and stability of the measurement.

*Dynamic range determines the ability to measure strongly attenuating samples.”

We considered performing FDTD simulations to examine the THz power absorption by a single NW and their ensembles. However, it is challenging. Firstly, such simulation requires the input of accurate conductivity and dielectric properties of the studied material. Our horizontal InAs NWs are a new type of material which hasn't been completely characterised, although we have developed a well-controlled growth technique to synthesise them. As mentioned in Pages 6 and 7 in the Supplementary Information, there is a lack of proper characterisation tools to study the single/enssembled NWs in terms of their optical/electrical/optoelectronic properties at the moment. We tried, alternatively, to use the dielectric properties of bulk InAs in our simulations. The simulated results do not match our experimental data well. Thus, we are developing near-field THz measurements on individual semiconductor nanostructures, which has the potential to measure the conductivity and dielectric properties of single NWs and will be

demonstrated in our follow-up work. Secondly, considering the large simulation domain size in the THz regime and the small feature of a single NW (~ 50 nm in thickness), such simulation is memory-consuming and inefficient. The optimisation of the simulation parameters is thus demanding. We believe that the work presented in this manuscript will inspire more efforts in the THz simulation field.

I would like to agree with the conclusion of the authors that the proposed novel devices under the similar excitation conditions might become competitive to ion-implanted bulk InP receivers, however, in order to support the claim that the proposed technological approach can become a break-through technology, a comparison with state-of-the-art photoswitches outside of nanowire research subfield is mandatory. That might also require revisiting the radiation coupling scheme. To my knowledge, a front-side coupling of THz radiation to a broadband antenna as it was employed by the authors is inferior to a well-aligned substrate lens configuration which is routinely implemented in commercial photoconductive THz emitters and receivers.

We thank the referee for giving such constructive comments. We agree that the use of hyperhemispherical lenses to improve radiation coupling is well established in the field of single-element photoconductive THz emitters and receivers. However this engineering enhancement which comes from improved THz radiation coupling would be very similar for both a bulk/thin-film InP:Fe⁺ detector and a NW detector. We chose not use substrate lenses for NW receivers or the reference InP:Fe⁺ detector for two reasons 1) We are aiming for an application in a large-area focal plane array for which a hyperhemispherical lens is inappropriate; 2) We find that the signal from any photoconductive THz receiver is very sensitive to the alignment of the hyperhemispherical lenses, meaning an additional variable when trying to compare receivers. We plan to further develop single element detectors based on our technology and compare benchmark against commercial detectors using industry standard fibre lasers, however a commercially optimized THz receiver was not the aim of this study on a very recently developed and novel nanomaterial.

We claimed our work as significant for the field of THz science and technology as well as the semiconductor nanowire community for the following reasons. 1) our approach enables scalable manufacturing of horizontal NW THz receivers in a fast and effective way. This used to be a big challenge that had hindered the practical application of NW receivers for years. In terms of an individual NW receiver, it is of importance to allow a high degree of flexibility to scale up/down the device dimensions to meet any application requirements, for example, for use in free-space THz systems or for near-field THz detection. Our approach addressed this challenge, and we provided proof-of-concept examples by demonstrating 1NW, 5NW and 20 NW receivers in this work. In terms of NW receiver array, we proposed a feasible route with this approach to develop high-speed broadband kilo-pixel focal-plane receive arrays (THz cameras) for spectroscopic imaging. This is highly demanded in THz imaging as this field is still under development considering the current commercially available THz cameras either having a limited detectable spectral range (<1 THz) or having a slow response speed (100-1000 ms) that imposes difficulties for real-time imaging. 2) With this approach, we show additional freedom to tune the NW properties by facet engineering to enable direct sampling without the need of additional material treatments after growth, e.g. ion-implantation/surface passivation treatment. Thus, this approach offers a new path to design photoconductive materials for optimal THz detection. Techniques such as low-temperature growth and ion implantation were no longer the only two options in this field. We believe our findings here will trigger massive attention and follow-up activities in both the fields of NW research science and THz

science and technology. 3) the most significant advantage of using NWs for THz detection is to allow the detection of full polarisation state of the THz radiation, which could lead to advanced THz applications such as THz polarimetry, THz polarisation imaging and THz polarisation communication. In particular, this approach will boost the development of polarisation-sensitive THz cameras that have not been demonstrated yet which could revolutionise the entire THz research. To emphasise this viewpoint, we have added some comments in the Conclusion.

*Changes page 11, addition of the following text:

“Significantly, in contrast to semiconducting films, NWs are intrinsically polarisation-sensitive to THz radiation [39] and naturally form low dark current, electrically isolated THz receiver elements [41]. These features make our scalable InAs NW technology promising for the development of polarisation-sensitive THz cameras thereby enabling future THz polarisation imaging applications.”

REVIEWERS' COMMENTS

Reviewer #1 (Remarks to the Author):

The authors took into account the comments and adjusted the text of the article and its supplement accordingly. I think the manuscript is suitable for publication.

Reviewer #2 (Remarks to the Author):

I want to thank the authors for providing detailed response to my comments and the corresponding revisions.

The authors have addressed my first comment (Are the GaAs nanoridges necessary...) actually by their response to the second comment of Reviewer 1. The revision regarding the impurities or contamination is less convincing, as they may also affect the growth of GaAs even as a buffer.

Regarding my second comment (etching vs. growth), I agree with the authors that etching does not allow for facet engineering. As I referred to InAs film (e.g., grown by MBE on GaAs), so the authors do not need to add "it would also be challenging to isolate those nanowires optically and electrically from the rest of the substrate."

Additionally, I feel the comparison shown in the new Table S1 is a little unfair between direct (apple) and integrating (orange) types of THz detection. More meaningful comparison may be between the direct (apple) and the derivative of the integrating (now also apple) THz detection.

My above points are rather minor, and now I can support the publication of this work.

Reviewer #3 (Remarks to the Author):

The authors appropriately addressed my questions raised in the first review. Therefore, I would like to suggest accepting the manuscript for publication in the present form.

REVIEWERS' COMMENTS**Reviewer #1** (Remarks to the Author):

The authors took into account the comments and adjusted the text of the article and its supplement accordingly. I think the manuscript is suitable for publication.

We thank the reviewer for their comment.

Reviewer #2 (Remarks to the Author):

I want to thank the authors for providing detailed response to my comments and the corresponding revisions.

The authors have addressed my first comment (Are the GaAs nanoridges necessary...) actually by their response to the second comment of Reviewer 1. The revision regarding the impurities or contamination is less convincing, as they may also affect the growth of GaAs even as a buffer.

We thank the reviewer for their comments. While it's true that impurities on the substrate surface can also affect the GaAs buffer layer growth, we nevertheless feel that the role of this buffer in separating the NW from the surface impurities is worth mentioning, so we have left the text unchanged.

Regarding my second comment (etching vs. growth), I agree with the authors that etching does not allow for facet engineering. As I referred to InAs film (e.g., grown by MBE on GaAs), so the authors do not need to add "it would also be challenging to isolate those nanowires optically and electrically from the rest of the substrate."

While a thin film of InAs would be easier to optically and electrically isolate from a GaAs substrate, special processing would still be needed to avoid damage from the etching process. We have therefore revised the text on page 12 again to highlight this.

*Changes page 11, addition of the following text:

"However, this approach does not allow for facet engineering [71] and requires special processing to avoid damage to the NW surface from the etching."

Additionally, I feel the comparison shown in the new Table S1 is a little unfair between direct (apple) and integrating (orange) types of THz detection. More meaningful comparison may be between the direct (apple) and the derivative of the integrating (now also apple) THz detection.

We agree with the referee and have now included a direct comparison of the dynamic range between the fully processed spectra data (i.e. "apple") as a new column labeled "Dynamic Range of spectral data" in an updated version of Table S1. We retain our original comparison between the raw (unprocessed)

data for completeness. Moreover, in the original Table S1, we inadvertently label 'single-faceted' as 'multi-faceted receivers' and multi-faceted' as 'single-faceted receivers'.

The lower dynamic range in the amplitude spectrum of the 5NW and 20NW samples is consistent with what is written in the main text, e.g. "Moreover, despite showing a hundred times lower measured signal strength, the multi-faceted 1NW receiver shows a comparable performance to a 'direct sampling' InP ..."

*New version of Table S1 in the SI with the changes described above:

Table S1. Detector performance comparison. Comparison of detection performance between 1NW, 5NW and 20NW receivers. (For signal-to-noise ratio of time-domain data, the signal is defined as the peak-to-peak current over one time-domain scan, and the noise is the standard deviation of the difference of two consecutive scans with identical parameters. Dynamic range of time-domain data is defined as the ratio of the peak-to-peak current over one time-domain scan to the standard deviation of the noise current in the absence of THz over the same scan. Dynamic range of spectral data is defined as the ratio of the maximum amplitude to the root mean square of the noise floor of the amplitude spectrum. Spectral response bandwidth is defined as the cut-off frequency at the noise floor of its spectral response.) All NW receivers have an identical antenna gap size of 1 μm and were optically gated with 1.5- μm laser pulses of 1.66 mJ/cm^2 fluence.

		Detector Type		
		Photoconductive Antenna (bow tie)		
Performance		1NW	5 NW	20 NW
NW Facet Type		Multi-faceted	Single-faceted	Single-faceted
Sampling Modes		Direct	Integrating	Integrating
Raw data in time domain					Signal-to-noise Ratio	6.0	22	28
	Dynamic Range of time-domain data	94	94	140
Converted spectral data in frequency domain					Dynamic Range of spectral data	580	130	230
	Spectral Response	3.0	3.0	3.0

	Bandwidth (THz)			
	3dB cut-off frequencies (THz)	0.14- 0.91	0.14-1.49	0.38-1.34

- *Signal-to-noise ratio determines the accuracy and stability of the measurement.
- *Dynamic range determines the ability to measure strongly attenuating samples.
- *The red dashed line in spectral data represents the original spectrum of THz source measured by a bulk ion-implanted InP receiver.

My above points are rather minor, and now I can support the publication of this work.

Reviewer #3 (Remarks to the Author):

The authors appropriately addressed my questions raised in the first review. Therefore, I would like to suggest accepting the manuscript for publication in the present form.

We thank the reviewer for their comment.